# Urine: A Pitfall for Molecular Detection of Toscana Virus? An Analytical Proof-of-Concept Study

**DOI:** 10.3390/v16010098

**Published:** 2024-01-08

**Authors:** Antonio Mori, Andrea Matucci, Elena Pomari, Silvia Accordini, Chiara Piubelli, Annalisa Donini, Lavinia Nicolini, Concetta Castilletti

**Affiliations:** Department of Infectious, Tropical Diseases and Microbiology, IRCCS Sacro Cuore Don Calabria Hospital, Negrar di Valpolicella, 37024 Verona, Italy; andrea.matucci@sacrocuore.it (A.M.); silvia.accordini@sacrocuore.it (S.A.); chiara.piubelli@sacrocuore.it (C.P.); annalisa.donini@sacrocuore.it (A.D.); lavinia.nicolini@sacrocuore.it (L.N.)

**Keywords:** Toscana virus, urine, real-time RT-PCR, analytical proof-of-concept study

## Abstract

Toscana virus (TOSV), a sandfly-borne virus, is an important etiological agent in human acute meningitis and meningoencephalitis in the Mediterranean area during the summer. However, the actual number of TOSV infections is underestimated. Laboratory confirmation is necessary because TOSV infection has overlapping clinical features with other neuro-invasive viral infections. Nowadays, the reference test for direct diagnosis in the acute phase of TOSV infection is the PCR based method for detecting TOSV in cerebrospinal fluid and/or plasma, serum, or blood. Although poorly employed, urine is another helpful biological matrix for TOSV detection. Urine is a matrix rich in PCR inhibitors that affect PCR efficiency; consequently, false negatives could be generated. To investigate the potential effect of urine PCR inhibitors on TOSV detection, we compared undiluted and diluted urine using 10-fold series of spiked TOSV. The results showed a significant improvement in TOSV detection performance in diluted urine (1 TCID_50_ vs. 1 × 10^4^ TCID_50_ limit of detection and 101.35% vs. 129.62% efficiency, respectively, in diluted and undiluted urine). In conclusion, our data provide preliminary important insights into the use of diluted urine to limit the impact of the inhibitory effects of urine on the detection of TOSV in RT-PCR-based approaches.

## 1. Introduction

Toscana virus (TOSV) is an arthropod-borne virus belonging to the Sandfly fever Naples phlebovirus species included in the Phlebovirus genus (*Bunyaviridae* family). Like other Phleboviruses, TOSV is an enveloped virus characterized by a genome consisting of three single-stranded negative-sense RNA linear segments (S for small, M for medium, and L for large) encoding non-structural and structural proteins. Phylogenetic studies have reported the presence of three distinct genotypes designated as lineages A, B, and C (the latter of which was recently identified), which have geographically different distributions in the Mediterranean area [1]. TOSV transmission occurs through phlebotomine sand flies (such as P. perniciosus and P. perfiliewi). TOSV infection is usually asymptomatic or presents with mild symptoms, but because it is neuro-invasive, central nervous system (CNS) pathologies can occur. In fact, TOSV is a major cause of aseptic meningitis and encephalitis during the summer season in both residents of and travelers from endemic areas. Neuroinvasive disease, although self-resolving in most cases, can be sometimes severe [2].

Nowadays, the actual number of TOSV infections is underestimated as, generally, only severe infections requiring hospitalization, especially those with neurological symptoms, require specific investigations in order to obtain a confirmatory laboratory diagnosis. Regarding laboratory diagnostics, the detection of specific TOSV genomic regions using PCR-based molecular assays (mainly real time RT-PCR (RT-qPCR)) is the reference test used for direct diagnosis in the acute phase of TOSV infection [3]. Indeed, in the acute phase, molecular assays are preferable to serological tests, which have less sensitivity [4], especially in this stage and when there is strong cross-reactivity between phleboviruses, particularly those antigenically related to TOSV (such as other members of the Sandfly fever Naples virus species) and closely related viruses (e.g., Granada and Massilia viruses) [5].

The most commonly employed biological matrices for the detection of acute TOSV infections are cerebrospinal fluid (CSF) and/or plasma, serum, or blood. Urine can be also helpful for detecting phleboviruses [6]; furthermore, TOSV shedding concurs or follows viremia [7]. As noted, in complex biological matrices such as urine, the amplification efficiency of diagnostic PCR assays could be reduced by the presence of inhibitors [8,9]. Presently, urine is poorly utilized in the context of TOSV diagnosis, and little is known of the persistence, genome stability, and excretion kinetics of TOSV in this biological matrix. A potential pitfall of using urine for diagnostics is false negatives due to PCR reaction inhibitors, especially when the virus is present in low amounts.

In this analytical proof-of-concept study, in order to establish whether PCR inhibitors (such as urea and others) in urine affect TOSV genomic detection and to evaluate whether urine dilution results in improved TOSV detection, undiluted and diluted TOSV-spiked urine were tested. We employed the Trio TOSV real-time RT-PCR (Trio TOSV RT-qPCR) test developed by Thirion et al. [10], which is currently in use in our laboratory. This RT-qPCR test is a multi-region assay that groups the most widely used specific TOSV monoplex RT-qPCRs into a single reaction. Each TOSV monoplex RT-qPCR targets a different region located in the N gene on segment S [11,12,13]. The multi-region approach reduces the risk of false negatives due to the occurrence of mutations in regions complementary to the primers and probes and increases the power to detect all three lineages [10].

## 2. Materials and Methods

### 2.1. Virus Stock Preparation

Vero E6 cells (ATCC^®^ number CRL-1586™) were maintained in Eagle’s Minimum Essential Medium (EMEM) (Gibco, Life Technologies Europe, Milano, Italy) supplemented with 100 U/mL penicillin/streptomycin (Gibco) and 10% heat-inactivated foetal calf serum (FCS) (Gibco) at 37 °C and 5% CO_2_ in a humidified atmosphere. For virus stock preparation, Vero E6 cells were infected with TOSV lineage A (kindly provided by Professor Tiziana Lazzarotto, Regional Reference Centre for Microbiological Emergencies (CRREM), St. Orsola Malpighi University Hospital, University of Bologna, Bologna, Italy). Cell lysates were clarified, aliquoted, and stored at −80 °C until use. Virus titration was performed on the Vero E6 cell line using a limiting dilution assay; the titer was calculated using the method of Reed and Muench and expressed as the tissue culture infectious dose TCID_50_/mL [14]. The virus stock titer was 10^8^ TCID_50_/mL.

### 2.2. Spiking of Clinical Specimens

The urine samples used for the study were fully anonymized residual samples from routine annual checkups of health care workers (HCWs) with healthy results. Samples were stored in the Tropica Biobank at IRCCS Sacro Cuore Don Calabria Hospital. Undiluted and diluted urine samples (1:4 (*v*:*v*)) in phosphate buffer solution (PBS, pH 7.4, Gibco) were tested. The 1:4 dilution was considered the right compromise to dilute any inhibitory substances and maintain near sensitivity in subsequent molecular investigations [15,16]. The undiluted and diluted urine samples (1:4 (*v*:*v*)) in phosphate buffer solution (PBS, pH 7.4) were spiked with 10-fold serial dilutions of the titrated virus (1 × 10^7^ TCID_50_/mL) under biocontainment level 3 conditions (BSL3). All the aliquots were then inactivated using the EZ1^®^ DSP Virus Kit Lysis Buffer (QIAGEN, Milano, Italy) according to the manufacturer’s instructions. Spiked samples were stored at −80 °C before proceeding to RNA extraction.

### 2.3. RNA Extraction and Viral Genome Detection

After virus spiking, the total RNA was extracted from 200 μL of the 10-fold series of 10 logs of undiluted and diluted TOSV-spiked urine samples using the EZ1^®^ DSP Virus Kit on the EZ1 Advanced XL automated platform (QIAGEN, Milano, Italy) according to the manufacturer’s instructions. RNA was eluted in a final volume of 90 μL and analyzed in quadruplicate using the Trio TOSV RT-qPCR [10]. The assay was carried out in 20 μL with 5 μL of RNA, 5 μL of Reliance One-Step Multiplex Supermix 4X (Biorad, Hercules, CA, USA), primers and probes at the concentrations reported in Thirion et al. [10], and 0.5 μL of BSA 5 mg/mL as a PCR facilitator. The reaction was performed in a BioRad CFX96TM thermal cycler (Biorad) with the following cycling conditions: 50 °C for 10 min; 95 °C for 10 min; 45 cycles of 95 °C for 10 s; 60 °C for 30 s.

## 3. Results

To evaluate the effect of PCR inhibitors in urine, three different parameters were considered: (a) the limit of detection (LoD), (b) the difference between the Ct values calculated for diluted and undiluted urine for each dilution, and (c) the Trio TOSV RT-qPCR efficiency. The threshold was set at 200 for all Trio TOSV RT-qPCR analyses. The LoD was calculated as the last dilution with all four replicates being detected as TOSV-positive (100% positivity hit score) by the Trio TOSV RT-qPCR [10]. The LoDs were 1 × 10^4^ TCID_50_ for undiluted urine and 10 TCID_50_ for diluted urine. However, the Trio TOSV RT-qPCR detected the TOSV viral genome in three out of four replicates at 1 TCID_50_ for diluted urine (Table 1). The average value calculated for the difference between Ct values was 7.83 (SD ± 0.61). In undiluted urine, the Trio TOSV RT-qPCR reported a slope with a standard curve of −2.77 and an efficiency value higher than 120% (129.62%) (Figure 1). When efficiency is greater than 120%, a possible cause could be the influence of inhibitors on the RT-qPCR performance [17,18,19]. In 1:4 diluted urine, the slope was −3.29 and had an efficiency value of 101.35% (Figure 1).

These results demonstrated that, even in the presence of a PCR facilitator (i.e., BSA), the performance of the Trio TOSV RT-qPCR and the detection of TOSV in undiluted urine was significantly reduced compared to diluted urine.

## 4. Discussion

In the Mediterranean endemic area, TOSV is one of the most prevalent viruses associated with febrile illness and neuro-invasive infections during the summer season [5,20]. In Italy, TOSV is an important etiologic agent of meningitis in north–central regions [21,22]. Unfortunately, the number of reported cases is largely underestimated due to two main reasons: (a) TOSV’s asymptomatic or mild manifestations [23] and (b) a lack of knowledge among physicians about its potential to cause CNS diseases [1].

Three genetic lineages are currently known (lineages A, B, and C), with different distribution areas. Specifically, lineage A was detected in Italy, France, Turkey, Algeria, and Tunisia, whereas lineage B was detected in Portugal, Spain, France, Morocco, Croatia, and Turkey and lineage C was detected in Croatia and Greece [1]. Moreover, the co-circulation of two lineages within the same geographical area has also been reported [1]. Because of its continuing circulation, TOSV is enlarging its distribution area, consequently making it a potential emerging pathogen to be monitored.

In general, direct diagnosis in the acute phase of TOSV infection relies mainly on RT-qPCR testing of CSF, blood, plasma, and serum. The genomic RNA of TOSV [7] and other phleboviruses has also been detected in urine [6]. However, while urine is an ideal non-invasive sample type in addition or complementary to CSF and blood sampling, this biological matrix is poorly used in the context of TOSV diagnosis. In urine, TOSV shedding seems to coincide with or follow viremia, although the precise duration of the excretion of TOSV is not known [7].

Urine is a complex biological matrix rich in urea, which is an important critical component because it denatures polymerase. In addition, various substances (such as beta-hCG and crystals) also affect PCR efficiency [8]. Therefore, the question remains whether negative results are due to the presence of PCR inhibitors limiting TOSV detection or are related to the biological cycle and maintenance of TOSV in urine. At present, little is reported in the literature concerning TOSV and urine [24,25,26,27].

To elucidate the effect of urine PCR inhibitors on TOSV genomic RNA detection, we conducted an analytical proof-of-concept study comparing the TOSV detection performance of RT-qPCR assays in undiluted versus diluted urine. For this study, we used the Trio TOSV RT-qPCR developed by Thirion et al. [10], which is currently used in our laboratory. The assay is based on a multi-region approach combining three different monoplex RT-qPCRs in a single reaction [11,12,13]. In addition, the included assay developed by Pèrez-Ruiz et al. is potentially able to detect lineage C [1,13]. Another point in favor of the Trio TOSV RT-qPCR is its good sensitivity to lineages A and B of TOSV.

The results showed a significant improvement in TOSV detection in diluted urine, which was demonstrated by a gain of four magnitudes in the LoD values (1 TCID_50_ vs. 1 × 10^4^ TCID_50_, respectively) and a lowering on average of about eight Ct values calculated in diluted versus undiluted urine. There was also a marked improvement in the efficiency of RT-qPCR TOSV RNA detection in diluted urine (101.35% vs. 129.62% efficiency, respectively, in diluted and undiluted urine). For molecular diagnostic assays, the LoD is generally considered the lowest concentration of the target that can be detected in ≥95% of repeat measurements, is a measure of analytic sensitivity, and can be reported in several formats. It should be noted that, despite the fact that LoDs for the most approved clinical assays are commonly expressed as TCID_50_/mL. [28], other quantification methods, such as copies/mL, are also reliable, and could be more precise in LoD comparisons between assays [29]. The LoD can be evaluated in different ways (e.g., 10-fold dilution series or probit analysis) according to the intended use. For our study, we applied the 10-fold dilution series.

One shortcoming of our study is the lack of testing of lineages B and C. In addition, few clinical cases have been found in Veneto because it is not currently an endemic region; indeed the first positive cases have only been detected in the last two years [30]. Moreover, it has been difficult to obtain positive urine samples from other Italian referral laboratories because urine has only recently been considered a good matrix for the diagnosis of TOSV. As a consequence of this difficulty, clinical specimens were not included in the present study.

Another point to note is that the present study was conducted using an extraction method and molecular detection assay currently in use in our laboratory, meaning the results might vary depending on the different nucleic acid extraction methods and/or molecular assays used, suggesting the need to perform an analytical test in each laboratory. 

To the best of our knowledge, this is the first paper evaluating the impact of urine inhibition on the detection of TOSV genomic sequences in this biological matrix. Based on our data, the use of testing on diluted and undiluted urine could be recommended in suspected cases, especially when no other samples are available for molecular diagnosis during the early stage of infection. Moreover, the analysis of an internal amplification control is suggested to avoid the pitfall of false negatives.

Sadly, although TOSV plays an important role in causing CNS infections, this phlebovirus remains a neglected human pathogen that is rarely included in diagnostic algorithms for CNS infections and febrile illness during the summer period. Little is known about the role of urine as a potential source of dissemination and infection. In a recent study, Matusali et al. [31] detected the presence of TOSV RNA in both acellular and cellular fractions of semen and isolated infectious TOSV from seminal plasma, highlighting the potential of sexual transmission as an alternative route of viral spread. Therefore, further investigations may be useful and necessary.

In conclusion, although we are aware of the lack of studies on clinical specimens, we believe the results of our proof-of-concept study are useful to point out this potential pitfall for detecting TOSV. Hence, the inclusion of diluted urine (in addition to undiluted urine, plasma, and CSF) in the diagnostic procedures for suspected cases might improve the detection of TOSV viral RNA. However, to evaluate the utility and impact of using diluted urine as a routine specimen, clinical studies are needed. At present, urine remains a biological matrix to be considered in addition to or as a complement to CSF and blood sampling.

## Figures and Tables

**Figure 1 viruses-16-00098-f001:**
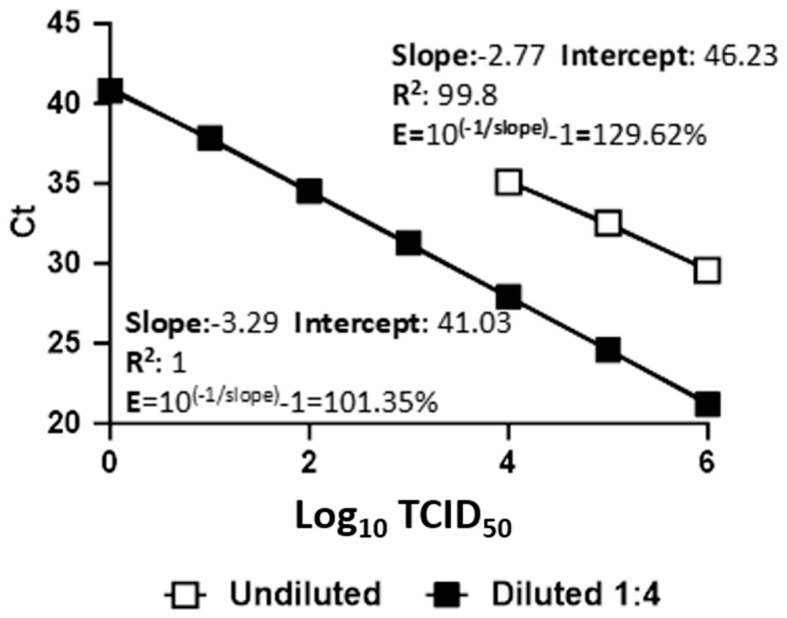
Comparison of the standard curves, correlation coefficients (R^2^), and efficiency (E) between undiluted and diluted urine. Viral RNA extracted from 10-fold dilutions in spiked undiluted (white square) and diluted urine (black square) was analyzed. Each plot represents the means of four replicate amplifications of each dilution. The x-axis shows the virus titration expressed in TCID_50_. The y-axis shows the Ct values.

**Table 1 viruses-16-00098-t001:** Analytical sensitivity (LoD) in undiluted and diluted urine.

TCID_50_	Undiluted Urine (Mean Ct ± SD ^3^)	Undiluted Urine Hit Score	Diluted Urine (Mean Ct ± SD ^3^)	Diluted Urine Hit Score	Undiluted-Diluted Mean Ct (ΔCt)
10^6^	29.57 (±0.25)	4/4	21.20 (±0.09)	4/4	8.37
10^5^	32.54 (±0.35)	4/4	24.59 (±0.04)	4/4	7.95
10^4^	**35.10** ^1^ (±0.36)	4/4	27.93 (±0.20)	4/4	7.17
10^3^	Nd ^2^	0/4	31.28 (±0.11)	4/4	
10^2^	nd	0/4	34.54 (±0.20)	4/4	
10^1^	nd	0/4	**37.87** ^1^ (±0.69)	4/4	
1	nd	0/4	40.81 (±1.01)	3/4	
0.1	nd	0/4	nd	0/4	
0.01	nd	0/4	nd	0/4	
0.001	nd	0/4	nd	0/4	

^1^ Limit of detection (LoD); ^2^ nd: not detected; ^3^ SD: standard deviation.

## Data Availability

All the data presented in this study are available in the article here.

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
