# Peer review of "Urine: A Pitfall for Molecular Detection of Toscana Virus? An Analytical Proof-of-Concept Study"

_viruses, 2024, doi:10.3390/v16010098_

Round 1

Reviewer 1 Report

Comments and Suggestions for Authors

Authors described urine as a relevant biological material for toscana virus ARN detection. They mentionned a 10 fold dilution of urine necessary for the best sentivity.

The paper is easy to read and well constructed. It should be noted, however, that this work is based on the use of a given extraction and PCR kit.

Readers who wish use the results of this study for their practice cannot expect to find the same conclusions with other kits. The authors should emphasize this fact.

Comments on the Quality of English Language

English is not my native language; I am not able to really comment the quality of english language even if the paper is highly redible.

Author Response

We thank the reviewer for the comment. In particular, we agree that the results depend on the performance of the extraction and purification approach as well as the molecular assays used. As suggested, we added now in the text that we applied extraction method and TOSV qRT-PCR currently in use in our laboratory and therefore the results might vary depending on different nucleic acid extraction methods and/or molecular assays applied. Furthermore, we suggested the need to perform an analytical test in each laboratory. See Discussion.

Reviewer 2 Report

Comments and Suggestions for Authors

The Article is a brief research report examining whether urine is a good specimen for detecting Toscana virus (TOSV) using PCR-based methods. TOSV is an important cause of aseptic meningitis and encephalitis in the Mediterranean region during summer. However, TOSV infections are likely underestimated due to asymptomatic/mild cases and lack of testing.

The authors spike undiluted and diluted urine samples with serial dilutions of a TOSV stock. They then extract RNA and test for TOSV using a TOSV-specific real-time RT-PCR assay targeting three genomic regions.

They find much better TOSV detection performance in diluted urine compared to undiluted urine:

  • 1 TCID50 vs 1x10^4 TCID50 limit of detection
  • 101.35% vs 129.62% PCR efficiency
  • Average 8 Ct value difference between diluted and undiluted urine

The authors conclude that urine contains PCR inhibitors that negatively impact TOSV detection. Diluting the urine helps reduce these inhibitory effects and significantly improves assay sensitivity and efficiency for detecting TOSV RNA. They recommend using diluted urine for TOSV molecular diagnostics.

The information presented in the paper is supported by the experiment performed and presented in a scientifically sound way, but it is very well known that urine inhibits PCR. And dilution of urine decreases the inhibition seen in clinical samples. This study could be done with any pathogen and has been published. If this study included clinical specimens that were missed or expanded on the time frame of shedding of virus from infected patients in conjunction with this information, then I would see the benefit of publishing this study.

Author Response

We thank the reviewer for the comment and suggestion. We agree it is well known that urine inhibits PCR, hence we wanted to conduct an analytical proof-of-concept study as we now specified in the title and in the text of the manuscript. Our study is aimed at providing preliminary experimental data related to the use of diluted urine in order to improve the sensitivity of TOSV detection by PCR based approach. We agree that these data should be extended on clinical samples but our region, Veneto, is not endemic and urine are a relatively recently introduced matrix in the diagnosis of TOSV infection, for this reason it is really difficult to obtain an adequate number of positive urine specimens from other Italian regional referral laboratories (even in endemic areas). We now expanded the description of this study limitation in the text (see Discussion).

Reviewer 3 Report

Comments and Suggestions for Authors

Manuscript ID: viruses-2758223

Title: Urine: A Pitfall for Molecular Detection of Toscana Virus?

The manuscript describes the use of PCR in urine for detection of Toscana virus. Urine is a tricky sample, due to presence of inhibitors, which can lead to false negative results. The study presents the improvements produced by diluting the urine. Though the improvements seem to be significant, the question remains, if urine samples are indeed useful for Toscana virus detection, as not a lot of evidence exists on detection of the virus in urine. A paired sample (blood, urine and eventually CSF) comparison should be attempted, to verify the usefulness of urine. Although the study indicates how to avoid false negatives in urine, the question still remains if it is an appropriate sample and at what time point in infection. This should be indicated in the discussion, and urine can at the moment not replace other preferred samples, but can only be an addition to the current testing protocol. The manuscript is well structured and the English language is very good.

Author Response

We thank the reviewer for the comment. We perfectly agree urine currently remain an additional or complement matrix compared with CSF and blood testing in the Toscana virus work-up. The aim of our study is to provide a message that the routine use in experimental and/or diagnostic procedures of paired diluted and undiluted urine could help to understand, without analytical bias, the duration of TOSV excretion and shedding, allowing so to answer the role of such matrix in TOSV infection and if urine samples are an appropriate matrix for Toscana virus detection TOSV. Indeed, the use of diluted urine, in addition to undiluted, could improve the sensitivity of TOSV detection by PCR in diagnostic procedures, especially in those situations where specimens for viral diagnostics of suspected cases are not available during the early stages of the infection. We now specified that our study is an analytical proof-of concept aimed at providing preliminary data related specifically to the dilution of urine. As suggested, we modified now the text speculating the use of diluted urine in addition to undiluted urine and to the conventional specimens CSF/blood. Moreover, we highlighted some study limitations (see Discussion).

Reviewer 4 Report

Comments and Suggestions for Authors

This is a short report on potential inhibitory effects present in urine in the detection of Toscana virus

although of interest, this paper is more the start of a project. You have now proven that you can detect cultured virus in relative high dose in diluted urine. But for the interest of the reader this must be completed with clinical samples and more background on the feasibility of a urine PCR. what would be the idea to chose urine of liquor or plasma? Especially with the presence of inhibitory factors. 

Did you try a cohort (I see one sentence in the discussion stating that your lab is in a non endemic area) but there must be reference laboratory samples. Although technically sound this paper is merely a report for a master thesis and the start of a project. Not the finished product (a paper) 

Author Response

We agree with the reviewer that this is like a start of a project. Indeed, we now specified in the title and in the text of the manuscript that our study is an analytical proof-of concept. Moreover, as described now in the paper, urine is currently a biological matrix to be considered in addition or as a complement to the conventional CSF and blood sampling for TOSV detection because urine can be an important resource when specimens for viral diagnostics of suspected cases are not available during the early stages of the infection. Our aim is to provide preliminary data related specifically to the dilution of urine in order to improve the sensitivity of TOSV detection by PCR based approach. Our study provides first message that the use of paired diluted and undiluted urine could help the TOSV detection in such biological matrix. We agree that these data should be extended on clinical samples but our region Veneto is not endemic and positive urine specimens are very difficult or impossible to obtain from other Italian laboratories (even in endemic areas), being urine a rarely used matrix for TOSV detection. We now expanded the description of this study limitation in the text (see Discussion).

Round 2

Reviewer 2 Report

Comments and Suggestions for Authors

No further changes needed

Author Response

We thank the reviewer

Reviewer 4 Report

Comments and Suggestions for Authors

-

Author Response

We thank the reviewer